# Non-approximate Inference
# for Collective Graphical Models on Path Graphs
# via Discrete Difference of Convex Algorithm

**Yasunori Akagi**
NTT Human Informatics Laboratories,
NTT Corporation
yasunori.akagi.cu@hco.ntt.co.jp

**Naoki Marumo**
NTT Communication Science Laboratories,
NTT Corporation
naoki.marumo.ec@hco.ntt.co.jp

**Hideaki Kim**
NTT Human Informatics Laboratories,
NTT Corporation
hideaki.kin.cn@hco.ntt.co.jp

**Takeshi Kurashima**
NTT Human Informatics Laboratories,
NTT Corporation
takeshi.kurashima.uf@hco.ntt.co.jp

**Hiroyuki Toda**
NTT Human Informatics Laboratories,
NTT Corporation
hiroyuki.toda.xb@hco.ntt.co.jp

## Abstract

The importance of aggregated count data, which is calculated from the data of multiple individuals, continues to increase. Collective Graphical Model (CGM) is a probabilistic approach to the analysis of aggregated data. One of the most important operations in CGM is maximum a posteriori (MAP) inference of unobserved variables under given observations. Because the MAP inference problem for general CGMs has been shown to be NP-hard, an approach that solves an approximate problem has been proposed. However, this approach has two major drawbacks. First, the quality of the solution deteriorates when the values in the count tables are small, because the approximation becomes inaccurate. Second, since continuous relaxation is applied, the integrality constraints of the output are violated. To resolve these problems, this paper proposes a new method for MAP inference for CGMs on path graphs. Our method is based on the Difference of Convex Algorithm (DCA), which is a general methodology to minimize a function represented as the sum of a convex function and a concave function. In our algorithm, important subroutines in DCA can be efficiently calculated by minimum convex cost flow algorithms. Experiments show that the proposed method outputs higher quality solutions than the conventional approach.

## 1 Introduction

In recent years, the importance of aggregated count data, which is calculated from the data of multiple individuals, has been increasing [21, 27]. Although technologies for acquiring individual data such as sensors and GPS have greatly advanced, it is still very difficult to handle individual data due to privacy concerns and the difficulty of tracking individuals. However, there are many situations where data aggregated from multiple individuals can be obtained and utilized easily. For example, Mobile Spatial Statistics [22], which is the hourly population data of fixed-size square grids calculated from

35th Conference on Neural Information Processing Systems (NeurIPS 2021).

cell phone network data in Japan, are available for purchase; such data is being used for disaster prevention and urban planning [20]. In traffic networks, traffic volume data at each point can be obtained more easily by sensors or cameras than the trajectories of individual cars, and the data is useful for managing traffic congestion [9, 26].

Collective Graphical Model (CGM) [17] is a probabilistic model to describe aggregated statistics of a sample drawn from a graphical model. CGM makes it possible to conduct various practical tasks on aggregated count data, such as estimating movements from population snapshots, parameter learning of the underlying graphical model, and interpolation and denoising of count tables. Particularly, the case where the underlying graph is a path graph is important because CGMs on path graphs can treat time series data in which the states of interest follow Markov chains. In fact, most of the real-world applications of CGMs utilize CGMs on path graphs to represent the collective movement of humans and animals [2, 4, 19]. Detailed analyses of time series of collective people movements from limited observations would be useful for controlling people flow to avoid congestion and to maintain social distancing in urban spaces.

One of the most important operations in CGM is maximum a posteriori (MAP) inference. MAP inference is the discrete (combinatorial) optimization problem of finding an assignment of unobserved variables that maximizes the posterior probability under given observations. MAP inference makes it possible to interpolate missing values of aggregated data and to estimate more detailed information that lies behind the observations. Unfortunately, MAP inference for general CGMs has been shown to be NP-hard [16] and thus is difficult to solve exactly and efficiently. Therefore, an alternative approach that solves an approximate problem, which is derived by applying Stirling's approximation and continuous relaxation, has been proposed [16]. Subsequent studies have focused on solving this approximate problem efficiently [12, 18, 19, 23].

However, there are inherent problems with this approach of solving the approximate problem. First, this approach tends to output a solution with a low posterior probability when the values in the count tables are small, because Stirling's approximation, $\log x! \approx x \log x - x$, is inaccurate when $x$ is small. Such a situation frequently occurs when the number of values that each variable in the graphical model takes is large, or when the sample size is small. Second, since continuous relaxation is applied, the integrality constraints of count table values are violated in the output. As a result, values that should be integers (e.g., the number of people) are no longer integers, which not only reduces interpretability, but also makes the output less sparse, resulting in high memory consumption to maintain the output. It is possible to obtain integer-valued results by rounding the output, but this rounding process destroys the sum constraints among the estimated counts, e.g., the sum of the count table values at each node may not match the sample size.

To resolve these issues, in this paper, we propose a new method for MAP inference for CGMs on path graphs. We first show that the objective function of the problem can be expressed as the sum of univariate discrete convex functions and discrete concave functions. Based on this expression, we utilize the idea of the Difference of Convex Algorithm (DCA) [6]. DCA is a framework to minimize a function expressed as the sum of a convex function and a concave function. In DCA, a solution is obtained by repeatedly minimizing a surrogate function that upper-bounds the objective function, and the objective function value decreases monotonically in each iteration. In addition, the algorithm terminates in a finite number of iterations in our case since the variables are discrete not continuous.

The key to make the DCA-based algorithm efficient is a fast minimization algorithm for the surrogate function. Because the feasible region of our problem is limited to integer lattice points, continuous optimization methods such as the gradient descent, which are usually used in DCAs, cannot be applied to minimize our surrogate function. Instead, we utilize the special structure of path graphs; it enables us to formulate the minimization problem of the surrogate function as a combinatorial optimization problem called the *minimum convex cost flow* problem. Fast algorithms for the minimum convex cost flow problem are known and we can minimize the surrogate function efficiently by using these algorithms.

The proposed method has several practical advantages. First, since the proposed method does not use Stirling's approximation, it offers an accurate inference even when the values in the count tables are small. This makes it possible to output solutions with much higher posterior probability than the approximation-based approach. Second, because the proposed method does not apply continuous relaxation, the obtained solution is guaranteed to be integer-valued, which results in sparse and interpretable outputs. In Section 5, we show results gained from synthetic and real-world datasets;

they indicate that the proposed method outputs higher quality solutions than the existing approach. We show that the superiority of the proposed method is much greater when the sample size is not very large or the number of states on nodes in the graphical model is large.

## 2 Collective Graphical Models (CGMs)

Collective Graphical Model (CGM) is a probabilistic generative model that describes the distributions of aggregated statistics of a sample drawn from a certain graphical model [17]. Let $G = (V, E)$ be an undirected tree graph (i.e., a connected graph with no cycles). We consider a pairwise graphical model over discrete random variable $\boldsymbol{X} \coloneqq (X_u)_{u \in V}$ defined by $\Pr(\boldsymbol{X} = \boldsymbol{x}) = \frac{1}{Z} \prod_{(u,v) \in E} \phi_{uv}(x_u, x_v)$, where $\phi_{uv}(x_u, x_v)$ is a local potential function on edge $(u, v)$ and $Z \coloneqq \sum_{\boldsymbol{x}} \prod_{(u,v) \in E} \phi_{uv}(x_u, x_v)$ is the partition function for normalization. In this paper, we assume that $x_u$ takes values on the set $[R]$ for all $u \in V$, where $[k]$ denotes the set $\{1, 2, \ldots, k\}$ for a positive integer $k$.

We draw an ordered sample $(\boldsymbol{X}^{(1)}, \ldots, \boldsymbol{X}^{(M)})$ independently from the graphical model, where $M$ is the sample size. Let $\boldsymbol{n}_u \coloneqq (n_u(i))_{i \in [R]}$ and $\boldsymbol{n}_{uv} \coloneqq (n_{uv}(i,j))_{i,j \in [R]}$, where $n_u(i) \coloneqq |\{m \mid X_u^{(m)} = i\}|$ and $n_{uv}(i,j) \coloneqq |\{m \mid X_u^{(m)} = i, X_v^{(m)} = j\}|$. Each entry of $\boldsymbol{n}_u$ and $\boldsymbol{n}_{uv}$ is the number of occurrences of a particular variable setting. We call $(\boldsymbol{n}_u)_{u \in V}$ *node contingency table* and $(\boldsymbol{n}_{uv})_{(u,v) \in E}$ *edge contingency table*, and denote $\boldsymbol{n} \coloneqq ((\boldsymbol{n}_u)_{u \in V}, (\boldsymbol{n}_{uv})_{(u,v) \in E})$. We assume that observations $\boldsymbol{y} \coloneqq (\boldsymbol{y}_u)_{u \in V}$ are generated by adding noise to the node contingency table $(\boldsymbol{n}_u)_{u \in V}$, and the distribution of $\boldsymbol{y}$ is given by $\Pr(\boldsymbol{y}|\boldsymbol{n}) = \prod_{u \in V} \prod_{i \in [R]} p_{ui}(y_u(i)|n_u(i))$, where $p_{ui}$ is the noise distribution. An additional assumption is described below.

**Assumption 1.** *For $u \in V$ and $i \in [R]$, $\log p_{ui}(y|n)$ is a concave function in $n$.*

Assumption 1 is a quite common assumption in CGM studies [16, 19]. Commonly used noise distributions such as Gaussian distribution $p_{ui}(y_u(i)|n_u(i)) = \frac{1}{\sqrt{2\pi\sigma^2}} \exp\left(\frac{-(y_u(i)-n_u(i))^2}{2\sigma^2}\right)$ and Poisson distribution $p_{ui}(y_u(i)|n_u(i)) = n_u(i)^{y_u(i)}/y_u(i)! \cdot \exp(-n_u(i))$ satisfy Assumption 1.

The **MAP inference problem** for CGM is to find $\boldsymbol{n}$ that maximizes the posterior probability $\Pr(\boldsymbol{n}|\boldsymbol{y})$. Since $\Pr(\boldsymbol{n}|\boldsymbol{y}) = \Pr(\boldsymbol{n}, \boldsymbol{y})/\Pr(\boldsymbol{y})$ from Bayes' rule, it suffices to maximize the joint probability $\Pr(\boldsymbol{n}, \boldsymbol{y}) = \Pr(\boldsymbol{n}) \cdot \Pr(\boldsymbol{y}|\boldsymbol{n})$. $\Pr(\boldsymbol{n})$ is called CGM distribution and calculated as follows [19]:

$$\Pr(\boldsymbol{n}) = F(\boldsymbol{n}) \cdot \mathbb{I}(\boldsymbol{n} \in \mathbb{L}_M^{\mathbb{Z}}), \tag{1}$$

$$F(\boldsymbol{n}) \coloneqq \frac{M!}{Z^M} \cdot \frac{\prod_{u \in V} \prod_{i \in [R]} (n_u(i)!)^{\nu_u - 1}}{\prod_{(u,v) \in E} \prod_{i,j \in [R]} n_{uv}(i,j)!} \cdot \prod_{(u,v) \in E} \prod_{i,j \in [R]} \phi_{uv}(i,j)^{n_{uv}(i,j)}, \tag{2}$$

$$\mathbb{L}_M^{\mathbb{Z}} \coloneqq \left\{ \boldsymbol{n} \in \mathbb{Z}_{\geq 0}^{|V|R + |E|R^2} \,\middle|\, M = \sum_{i \in [R]} n_u(i) \; (u \in V), \right.$$

$$\left. n_u(i) = \sum_{j \in [R]} n_{uv}(i,j) \; ((u,v) \in E, \; i \in [R]) \right\}. \tag{3}$$

Here, $\mathbb{I}(\cdot)$ is the indicator function, $\nu_u$ is the degree of node $u$ in $G$, and $\mathbb{L}_M^{\mathbb{Z}}$ is the set of possible contingency tables. Using the above notations, the MAP inference problem can be written as

$$\min_{\boldsymbol{n} \in \mathbb{L}_M^{\mathbb{Z}}} \; -\log F(\boldsymbol{n}) - \log \Pr(\boldsymbol{y}|\boldsymbol{n}). \tag{4}$$

## 3 CGMs on Path Graphs

Hereafter, we focus on CGMs on path graphs, which is the main topic of this paper. Path graph $P_n$ is an undirected graph whose vertex set is $V = [N]$ and edge set is $E = \{(t, t+1) \mid t \in [N-1]\}$. A graphical model (not CGM) on path graph is the most basic graphical model that represents a time series generated by a Markov model; that is, the current state depends only on the previous state. A CGM on a path graph represents the distribution of aggregated statistics when there are many individuals whose state transition is determined by a Markov model. In the rest of this paper, we use the notation $n_{ti} \coloneqq n_t(i)$, $n_{tij} \coloneqq n_{t,t+1}(i,j)$, and $\phi_{tij} \coloneqq \phi_{t,t+1}(i,j)$ for simplicity. From (1)–(4),

the MAP inference problem for CGMs on path graphs can be written as follows:

$$\min_{\boldsymbol{n}} \quad \sum_{t=1}^{N-1} \sum_{i,j\in[R]} f_{tij}(n_{tij}) + \sum_{t=2}^{N-1} \sum_{i\in[R]} g(n_{ti}) + \sum_{t=1}^{N} \sum_{i\in[R]} h_{ti}(n_{ti}),$$

$$\text{s.t.} \quad \sum_{i\in[R]} n_{ti} = M \quad (t \in [N]), \quad \sum_{j\in[R]} n_{tij} = n_{ti} \quad (t \in [N-1], \ i \in [R]), \tag{5}$$

$$\sum_{i\in[R]} n_{tij} = n_{t+1,j} \quad (t \in [N-1], \ j \in [R]), \quad n_{tij}, n_{ti} \in \mathbb{Z}_{\geq 0},$$

where $f_{tij}(z) := \log z! - z \cdot \log \phi_{tij}, g(z) := -\log z!, h_{ti}(z) := -\log p_{ti}(y_{ti}|z)$. For the details of the derivation, please see Appendix.

We give an example of a CGM on a path graph which models human mobility. Consider that a space is divided into $R$ distinct areas and that $M$ people are moving around in the space. The random variable $X_t^{(m)}$ represents the area to which person $m$ belongs at time step $t$, and the time series $\boldsymbol{X}^{(m)} = \left(X_1^{(m)}, \ldots, X_N^{(m)}\right)$ is determined by the graphical model $p(\boldsymbol{x}) = \frac{1}{Z} \prod_{t=1}^{N-1} \phi_{tx_t x_{t+1}}$. Here, $\phi_{tij}$ is the affinity between two areas $i$ and $j$ at time step $t \to t+1$. $n_{ti}$ represents the number of people in area $i$ at time step $t$, and $n_{tij}$ represents the number of people who moved from area $i$ to $j$ at time step $t \to t+1$. We have noisy observations $y_{ti}$ for $t \in [N]$ and $i \in [R]$, which are generated by adding noise to $n_{ti}$. The MAP inference problem we want to solve is to find the true number of people of each area at each time step, $(n_{ti})_{t\in[N],i\in[R]}$, and the true number of people moving between each two areas, $(n_{tij})_{t\in[N-1],i,j\in[R]}$, with the highest posterior probability given the observation $(y_{ti})_{t\in[N],i\in[R]}$.

## 4 Proposed Method

### 4.1 Application of DCA

To solve problem (5), we propose utilizing the idea of the Difference of Convex Algorithm (DCA). Before describing our method, we review the core idea of DCA based on the description in [11].

DCA is a general framework to solve the minimization problem $\min_{\boldsymbol{n}\in D} \mathcal{P}(\boldsymbol{n}) = \mathcal{Q}(\boldsymbol{n}) + \mathcal{R}(\boldsymbol{n})$, where $\mathcal{Q}(\boldsymbol{n})$ is a convex function and $\mathcal{R}(\boldsymbol{n})$ is a concave function. DCA does this by using the following procedure to generate a feasible solution sequence $\boldsymbol{n}^{(1)}, \ldots, \boldsymbol{n}^{(s)}$ that satisfies $\mathcal{P}(\boldsymbol{n}^{(1)}) \geq \mathcal{P}(\boldsymbol{n}^{(2)}) \geq \cdots \geq \mathcal{P}(\boldsymbol{n}^{(s)})$. First, we choose an arbitrary feasible solution $\boldsymbol{n}^{(1)} \in D$. When we already have the sequence $\boldsymbol{n}^{(1)}, \ldots, \boldsymbol{n}^{(s)}$, we find a function $\bar{\mathcal{R}}^{(s)}(\boldsymbol{n})$ that satisfies the following three conditions: (i) $\bar{\mathcal{R}}^{(s)}(\boldsymbol{n}^{(s)}) = \mathcal{R}(\boldsymbol{n}^{(s)})$, (ii) $\bar{\mathcal{R}}^{(s)}(\boldsymbol{n}) \geq \mathcal{R}(\boldsymbol{n}) \ (\forall \boldsymbol{n} \in D)$, (iii) $\bar{\mathcal{P}}^{(s)}(\boldsymbol{n}) := \mathcal{Q}(\boldsymbol{n}) + \bar{\mathcal{R}}^{(s)}(\boldsymbol{n})$ can be minimized efficiently on $D$. Because $\mathcal{R}(\boldsymbol{n})$ is concave, by setting $\bar{\mathcal{R}}^{(s)}(\boldsymbol{n}) = \mathcal{R}(\boldsymbol{n}^{(s)}) + \nabla \mathcal{R}(\boldsymbol{n}^{(s)}) \cdot (\boldsymbol{n} - \boldsymbol{n}^{(s)})$, which is a linear approximation of $\mathcal{R}(\boldsymbol{n})$ at $\boldsymbol{n}^{(s)}$, conditions (i)–(ii) hold. Using this function, we get a new feasible solution by $\boldsymbol{n}^{(s+1)} = \arg\min_{\boldsymbol{n}} \bar{\mathcal{P}}^{(s)}(\boldsymbol{n})$. This can be done easily because condition (iii) holds. Then, because $\mathcal{P}(\boldsymbol{n}^{(s+1)}) \leq \bar{\mathcal{P}}^{(s)}(\boldsymbol{n}^{(s+1)}) \leq \bar{\mathcal{P}}^{(s)}(\boldsymbol{n}^{(s)}) = \mathcal{P}(\boldsymbol{n}^{(s)})$, we get $\mathcal{P}(\boldsymbol{n}^{(1)}) \geq \mathcal{P}(\boldsymbol{n}^{(2)}) \geq \cdots \geq \mathcal{P}(\boldsymbol{n}^{(s)}) \geq \mathcal{P}(\boldsymbol{n}^{(s+1)})$ by induction.

To apply the framework of DCA, the objective function must be expressed as the sum of convex and concave functions. The following proposition shows that our MAP inference problem in (5) has such a structure.

**Definition 1.** *A function $f : \mathbb{Z}_{\geq 0} \to \mathbb{R} \cup \{+\infty\}$ is called a discrete convex function when $f(z+2) + f(z) \geq 2 \cdot f(z+1)$ for all $z \in \mathbb{Z}_{\geq 0}$. If $-f$ is a discrete convex function, $f$ is called a discrete concave function.*

**Proposition 1.** *$f_{tij}$ is a discrete convex function. Under Assumption 1, $h_{ti}$ is a discrete convex function. $g$ is a discrete concave function.*

The proof is given in the Appendix. Hereafter, we set $\mathcal{Q}(\boldsymbol{n}) = \sum_{t=1}^{N-1} \sum_{i,j\in[R]} f_{tij}(n_{tij}) + \sum_{t=1}^{N} \sum_{i\in[R]} h_{ti}(n_{ti})$ and $\mathcal{R}(\boldsymbol{n}) = \sum_{t=2}^{N-1} \sum_{i\in[R]} g(n_{ti})$. Thanks to Proposition 1, we can apply DCA to our problem. The following proposition provides a function $\bar{\mathcal{R}}^{(s)}(\boldsymbol{n})$ that satisfies conditions (i) and (ii) required for DCA.

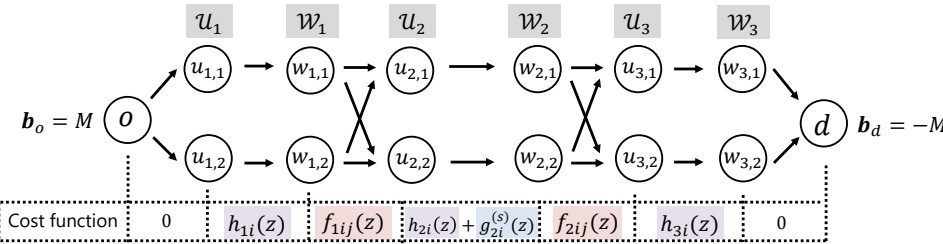

Figure 1: An example of the MCFP instance defined in Proposition 3 when $N = 3$ and $R = 2$.

**Proposition 2.** *Let $\bar{g}_{ti}^{(s)}(z) := -\log(n_{ti}^{(s)}!) + \alpha_{ti}^{(s)} \cdot (z - n_{ti}^{(s)})$, where $\alpha_{ti}^{(s)}$ is a real number which satisfies $-\log(n_{ti}^{(s)} + 1) \leq \alpha_{ti}^{(s)} \leq -\log n_{ti}^{(s)}$. Then, the function $\bar{\mathcal{R}}^{(s)}(\boldsymbol{n}) := \sum_{t=2}^{N-1} \sum_{i \in [R]} \bar{g}_{ti}^{(s)}(n_{ti})$ satisfies $\bar{\mathcal{R}}^{(s)}(\boldsymbol{n}^{(s)}) = \mathcal{R}(\boldsymbol{n}^{(s)})$ and $\bar{\mathcal{R}}^{(s)}(\boldsymbol{n}) \geq \mathcal{R}(\boldsymbol{n})$.*

Please see the Appendix for the proof. Intuitively, $\bar{g}_{ti}^{(s)}$ is a tangent of $g$ at $n_{ti}$.

## 4.2 Minimum Cost Flow Algorithm for the Subroutine

The most important and difficult part to derive an efficient DCA-based algorithm is designing efficient algorithms for the problem $\min_{\boldsymbol{n} \in D} \bar{\mathcal{P}}^{(s)}(\boldsymbol{n})$ (condition (iii)). To achieve this, we show that $\min_{\boldsymbol{n} \in D} \bar{\mathcal{P}}^{(s)}(\boldsymbol{n})$ can be formulated as the Minimum Convex Cost Flow Problem (C-MCFP), which is the efficiently solvable subclass of the Minimum Cost Flow Problem (MCFP). The (non-linear) MCFP is a combinatorial optimization problem on a directed graph $\mathcal{G} = (\mathcal{V}, \mathcal{E})$. Each node $i \in \mathcal{V}$ has a supply value $b_i \in \mathbb{Z}$, and each edge $(i, j) \in \mathcal{E}$ has a cost function $c_{ij} : \mathbb{Z}_{\geq 0} \to \mathbb{R} \cup \{+\infty\}$. MCFP is the problem of finding a minimum cost flow on $\mathcal{G}$ that satisfies the supply constraints at all nodes. MCFP can be described as follows:

$$\min_{\boldsymbol{z} \in \mathbb{Z}_{\geq 0}^{|\mathcal{E}|}} \sum_{(i,j) \in \mathcal{E}} c_{ij}(z_{ij}) \qquad \text{s.t.} \qquad \sum_{j:(i,j) \in \mathcal{E}} z_{ij} - \sum_{j:(j,i) \in \mathcal{E}} z_{ji} = b_i \quad (i \in \mathcal{V}).$$

Note that $\boldsymbol{z}$ takes only integer values (i.e., $\boldsymbol{z} \in \mathbb{Z}^{|\mathcal{E}|}$). A subclass of MCFP in which all cost functions are discrete convex functions (see Definition 1) is called the C-MCFP; it is known to be efficiently solvable [1]. The following proposition shows that the subproblem $\min_{\boldsymbol{n} \in D} \bar{\mathcal{P}}^{(s)}(\boldsymbol{n})$ can be formulated as a C-MCFP.

**Proposition 3.** *Define the MCFP instance as follows:*

- *the node set $\mathcal{V}$ is defined by $\mathcal{V} := \{o, d\} \cup (\cup_{t \in [N]} (\mathcal{U}_t \cup \mathcal{W}_t))$, where $\mathcal{U}_t := (u_{t,i})_{i \in [R]}$, $\mathcal{W}_t := (w_{t,i})_{i \in [R]}$,*
- *the edge set $\mathcal{E}$ consists of four types of edges,*
    - *edges $(o, u_{1,i}, 0)$ and $(w_{N,i}, d, 0)$ for $i \in [R]$,*
    - *edges $(u_{t,i}, w_{t,i}, h_{ti}(z))$ for $t = 1, N$ and $i \in [R]$,*
    - *edges $(u_{t,i}, w_{t,i}, \bar{g}_{ti}^{(s)}(z) + h_{ti}(z))$ for $t = 2, \ldots, N-1$ and $i \in [R]$,*
    - *edges $(w_{t,i}, u_{t+1,i}, f_{tij}(z))$ for $t \in [N-1]$ and $i, j \in [R]$,*
  *where $(u, v, c(z))$ represents a directed edge from node $u$ to node $v$ with cost function $c(z)$,*
- *the supply values $(b_i)_{i \in \mathcal{V}}$ are defined by $b_o = M$, $b_d = -M$, and $b_v = 0$ for $v \in \mathcal{V} \setminus \{o, d\}$.*

*Let $\boldsymbol{z}^*$ is an optimal solution of this MCFP instance, and define $\boldsymbol{n}^*$ by $n_{ti}^* := z_{u_{t,i}w_{t,i}}^*$ and $n_{tij}^* := z_{w_{t,i}u_{t+1,j}}^*$. Then, $\boldsymbol{n}^*$ is an optimal solution of the problem $\min_{\boldsymbol{n} \in D} \bar{\mathcal{P}}^{(s)}(\boldsymbol{n})$. Furthermore, the MCFP instance belongs to C-MCFP.*

The proof is given in the Appendix. Figure 1 illustrates an example of the MCFP instance defined in Proposition 3. This MCFP can be interpreted as the problem of finding a way to push $M$ flows from node $o$ to node $d$ with minimum cost. The above proposition enables us to solve the subproblem $\min_{\boldsymbol{n} \in D} \bar{\mathcal{P}}^{(s)}(\boldsymbol{n})$ efficiently by applying existing algorithms for C-MCFP.

### 4.3 Overall View of the Proposed Method and Time Complexity Analysis

From the above arguments, we can construct an efficient optimization algorithm, described in Algorithm 1, for the MAP inference problem (5). The algorithm is guaranteed to terminate after a finite number of iterations because $\mathcal{P}(\boldsymbol{n}^{(s)})$ monotonically decreases and $D$ is a finite set.

We analyze the time complexity of one iteration of the proposed method (Lines 3–5 in Algorithm 1). The computational bottleneck is solving C-MCFP in Line 3. There are several algorithms to solve

---

**Algorithm 1** DCA for problem (5)

1: $\boldsymbol{n}^{(1)} \leftarrow \boldsymbol{0}$
2: **for** $s = 1, 2, \ldots$ **do**
3: $\quad \boldsymbol{n}^{(s+1)} \leftarrow (\boldsymbol{n}^*$ defined in Proposition 3)
4: $\quad$ **if** $\mathcal{P}(\boldsymbol{n}^{(s)}) = \mathcal{P}(\boldsymbol{n}^{(s+1)})$ **then**
5: $\quad\quad$ **return** $\boldsymbol{n}^{(s)}$

---

C-MCFP and time complexity varies depending on which one is adopted. In this paper, we consider two typical methods, the Successive Shortest Path algorithm (SSP) and the Capacity Scaling algorithm (CS) [1].

SSP is an algorithm that successively augments unit flow along the shortest path from a supply node (i.e. $b_i > 0$) to a demand node (i.e. $b_i < 0$) in the *residual graph*, which is an auxiliary graph calculated from the current flow. Given a C-MCFP instance with graph $\mathcal{G} = (\mathcal{V}, \mathcal{E})$, the shortest path in the residual graph can be computed in $O(|\mathcal{E}| \log |\mathcal{V}|)$ time by Dijkstra's algorithm with a binary heap, and the augmentation of the flow can be done in $O(|\mathcal{E}|)$ time. The augmentation is performed $B := (\sum_{i \in \mathcal{V}} |b_i|)/2$ times totally, so the total time complexity is $O(B|\mathcal{E}| \log |\mathcal{V}|)$. CS resembles SSP, but it differs in that it tries to push a large amount of flow, rather than a unit amount of flow, in a single augmentation. In CS, the number of shortest path calculations and flow augmentations can be bounded $O(|\mathcal{E}| \log U)$ times, where $U := \max_{i \in \mathcal{V}} |b_i|$, so the total computational complexity is $O(|\mathcal{E}|^2 \log |\mathcal{V}| \log U)$.

Because $|\mathcal{V}| = O(NR)$, $|\mathcal{E}| = O(NR^2)$, $B = O(M)$ and $U = O(M)$ in our problem, the time complexity in one iteration is $O(MNR^2 \log(NR))$ when SSP is applied and $O(N^2 R^4 \log(NR) \log M)$ when CS is applied. This result implies that each method has its own advantages and disadvantages: SSP has small time complexity for $N$ and $R$, while CS has small time complexity for $M$. This difference is confirmed empirically in Section 5.1.

## 5 Experiments

We perform experiments to evaluate the effectiveness of the proposed method using synthetic and real-world instances. All experiments are conducted on a 64-bit macOS machine with Intel Core i7 CPUs and 16 GB of RAM. All algorithms are implemented in C++ (gcc 9.1.0 with -O3 option).

### 5.1 Synthetic Instances

**Settings.** We solve randomly generated synthetic instances of the MAP inference problem (5) and compare the attained objective values. We fix $N$ to 5 and vary the values of $R$ and $M$. The input observation $y_{ti}$ is independently drawn from uniform distribution on the set of integers $\{1, 2, \ldots, 2 \cdot \lfloor \frac{M}{R} \rfloor\}$. As the noise distribution, we use Gaussian distribution $p_{ti}(y_{ti}|n_{ti}) \propto \exp\left(-0.01 \cdot (y_{ti} - n_{ti})^2\right)$. We use two types of potential functions as follows. (1) **uniform**. $\phi_{tij}$ is independently drawn from uniform distribution on the set of integers $\{1, 2, \ldots, 10\}$. (2) **distance**. We set $\phi_{tij} = \frac{1}{|i-j+1|}$. This potential models the movement of individuals in one-dimensional space: the state indices $i$ and $j$ represent coordinates in the space, and the closer the two points are, the more likely are movements between them to occur.

**Proposed Method.** To construct surrogate functions in the proposed method, we can choose arbitrary $\alpha_{ti}^{(s)}$ which satisfies the condition $-\log(n_{ti}^{(s)} + 1) \le \alpha_{ti}^{(s)} \le -\log n_{ti}^{(s)}$ (see Proposition 2). To investigate the influence of the choice of $\alpha_{ti}^{(s)}$, we try three strategies to decide $\alpha_{ti}^{(s)}$: (1) $\alpha_{ti}^{(s)} = -\log(n_{ti}^{(s)})$, (2) $\alpha_{ti}^{(s)} = -\frac{1}{2}(\log(n_{ti}^{(s)}) + \log(n_{ti}^{(s)} + 1))$, (3) $\alpha_{ti}^{(s)} = -\log(n_{ti}^{(s)} + 1)$. We call them Proposed (L), Proposed (M), Proposed (R), respectively.

Table 1: Attained objective functions in synthetic instances. For each setting, we generated 10 instances and average values are shown. The smallest value is highlighted for each setting. U and D mean the "uniform" and "distance" potential settings, respectively.

| | $M$ | $10^1$ | | | $10^2$ | | | $10^3$ | | |
|---|---|---|---|---|---|---|---|---|---|---|
| | $R$ | 10 | 20 | 30 | 10 | 20 | 30 | 10 | 20 | 30 |
| U | Proposed (L) | **-9.97e+01** | **-8.90e+01** | **-8.74e+01** | **-1.11e+03** | **-1.19e+03** | **-1.22e+03** | **-1.07e+04** | **-1.31e+04** | **-1.40e+04** |
| | Proposed (M) | -9.81e+01 | -8.90e+01 | **-8.74e+01** | -1.11e+03 | -1.19e+03 | -1.22e+03 | -1.07e+04 | -1.31e+04 | -1.40e+04 |
| | Proposed (R) | -9.64e+01 | -8.76e+01 | **-8.74e+01** | -1.11e+03 | -1.18e+03 | -1.21e+03 | -1.07e+04 | -1.31e+04 | -1.40e+04 |
| | NLBP | -7.19e+01 | -7.01e+01 | -7.01e+01 | -1.08e+03 | -9.87e+02 | -9.02e+02 | -1.07e+04 | -1.30e+04 | -1.37e+04 |
| D | Proposed (L) | **3.35e-01** | **5.00e-01** | **5.00e-01** | **-5.48e+01** | **-3.03e+01** | **-1.18e+01** | **-5.83e+00** | **-9.06e+02** | **-1.01e+03** |
| | Proposed (M) | **3.35e-01** | **5.00e-01** | **5.00e-01** | -5.43e+01 | -2.91e+01 | -1.14e+01 | -5.82e+00 | -9.06e+02 | -1.01e+03 |
| | Proposed (R) | **3.35e-01** | **5.00e-01** | **5.00e-01** | -5.39e+01 | -2.89e+01 | -1.06e+01 | -5.80e+00 | -9.06e+02 | -1.01e+03 |
| | NLBP | 3.20e+01 | 4.56e+01 | 5.28e+01 | -1.38e+01 | 1.77e+02 | 3.25e+02 | 1.20e+00 | -8.02e+02 | -5.31e+02 |

**Compared Method.** As the compared method, we use Non-Linear Belief Propagation (NLBP) [19], which is a message-passing style algorithm to the solve approximate MAP inference problem derived by applying Stirling's approximation and continuous relaxation. Because the output of NLBP is not integer-valued and $\log(z!)$ is defined only if $z$ is an integer, we cannot calculate the objective function of (5) directly. To address this, we calculate it by replacing the term $\log(z!)$ by linear interpolation of $\log(\lfloor z \rfloor!)$ and $\log(\lceil z \rceil!)$, which is given by $(\lceil z \rceil - z) \cdot \log(\lfloor z \rfloor!) + (z - \lfloor z \rfloor) \cdot \log(\lceil z \rceil!)$. Note that although there are various algorithms to solve the approximate MAP inference problem (see Section 6.1), the objective function values attained by these algorithms are the same. This is because the approximate problem is a convex optimization problem [16].

**Comparison of attained objective values.** The results are shown in Table 1. We generated 10 instances for each parameter setting and determined the average of attained objective function values. Because the objective function $\mathcal{P}(\boldsymbol{n})$ is equal to $-\log \Pr(\boldsymbol{n}|\boldsymbol{y}) + \text{const.}$, $\mathcal{P}(\boldsymbol{n})$ takes both positive and negative values, and the difference of the objective function values is essential; when $\mathcal{P}(\boldsymbol{n}_1) - \mathcal{P}(\boldsymbol{n}_2) = \delta$, $\Pr(\boldsymbol{n}_1|\boldsymbol{y}) = \exp(-\delta) \cdot \Pr(\boldsymbol{n}_2|\boldsymbol{y})$ holds.

All the proposed methods consistently have smaller objective function values than the compared method. The difference tends to be large when $R$ is large and $M$ is small. This would be because small values appear in the contingency table more frequently when $R$ is large and $M$ is small, and the effect of the inaccuracy of Stirling's approximation becomes larger. Among the three proposed methods, there was not much difference in obtained objective function values although Proposed (L) was found to consistently achieve slightly smaller objective function values than others. This indicates that the proposed method is robust with respect to the choice of hyperparameters $\alpha_{ti}^{(s)}$.

**Characteristics of the output solution.** To compare the characteristics of solutions obtained by proposed (L) and NLBP, we solve an instance with $R = 20$, $M = 10^2$, and uniform potential by each method. Obtained edge contingency tables $n_{1ij}$ are shown in Figure 2 as heat maps. We also show the edge contingency table obtained by rounding the NLBP solution to integers. We observe that the proposed method outputs sparse solutions while the solutions by NLBP are blurred and contain a lot of non-zero elements. This difference is quantified by "sparsity", which is calculated by $1.0-$ (# of non-zero ($> 10^{-2}$) elements)/(# of elements): sparsity of the output of proposed (L) is 77%, while the sparsity of the output of NLBP is 0%. This is caused by its application of continuous relaxation and the inaccuracy of Stirling's approximation around 0. In the solution of NLBP (rounded), many near-zero values are rounded to 0, and constraints of the problem (5) are totally violated; for example, the sum of the edge contingency table values does not match the sample size. In additional experiments, we observed that the outputs of the three methods become closer as $M$ increases. For more details, please see the Appendix.

**Comparison of computation time.** We compare the computation time of each algorithm. As explained in Section 4.3, we can choose an arbitrary C-MCFP algorithm as the subroutine in the proposed method and the time complexity varies depending on the choice. We compare proposed (L) with SSP, proposed (L) with CS, and NLBP.

Figure 3 shows the relationship between input size and computation time, and Figure 4 shows the relationship between running time and objective function value. These results are consistent with the complexity analysis results in Section 4.3; SSP is efficient when $R$ is large but becomes inefficient

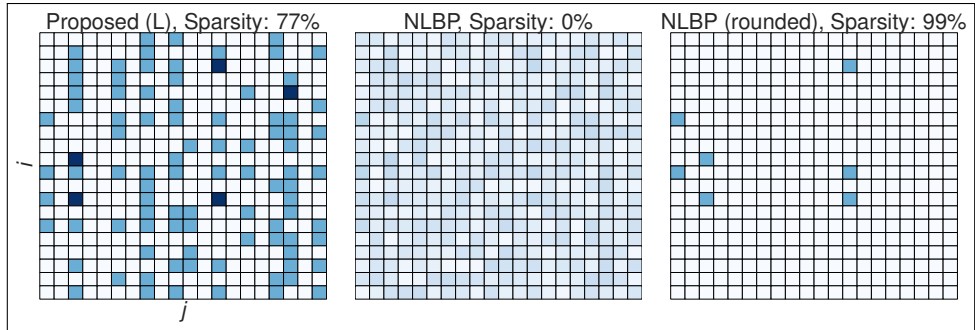

Figure 2: Comparison of solutions yielded by proposed method (L), NLBP, and NLBP (rounded). We solve an instance with $R = 20$, $M = 10^2$ and uniform potential. The obtained edge contingency table $n_{1ij}$ is presented as a matrix heatmap with the maximum value of color map 3.

Table 2: Attained objective function values and NAEs for real-world instances. For each setting, we generated 10 instances and averages are shown. The smallest value is highlighted.

| $M$ | | 100 | | 500 | | 1000 | |
|---|---|---|---|---|---|---|---|
| $R$ | | 56 | 208 | 56 | 208 | 56 | 208 |
| | Proposed (L) | **-3.02e+02** | **2.97e+02** | **-6.61e+03** | **-2.62e+03** | **-1.70e+04** | **-9.76e+03** |
| Obj. Val. | NLBP | 1.30e+03 | 3.24e+03 | -3.46+03 | 6.93e+03 | -1.34e+04 | 4.70e+03 |
| | NLBP (rounded) | - | - | - | - | - | - |
| | Proposed (L) | **0.690** | **0.441** | **1.002** | **0.829** | **1.073** | **0.956** |
| NAE | NLBP | 1.38 | 1.544 | 1.241 | 1.438 | 1.209 | 1.381 |
| | NLBP (rounded) | 0.877 | 0.870 | 1.079 | 0.907 | 1.128 | 0.991 |

when $M$ is large, and the converse is true for CS. The results also suggest that it is important to choose the algorithm depending on the size parameter of the input. The proposed method is not much worse than the existing method in terms of computation time by choosing an appropriate C-MCFP algorithm according to the size of the input. Appropriately chosen proposed methods attain the minimum of the existing method more quickly; proposed methods take a lot of time to achieve a smaller objective function value than the minimum of the existing method.

## 5.2 Real-world Instances

We conduct experiments using real-world population datasets. The datasets are generated from 8694 car trajectories collected by a car navigation application in the Greater Tokyo area, Japan [1]. We randomly sample $M$ ($M = 100, 500, 1000$) trajectories from this data and create aggregated population data of each area at fixed time intervals. The areas are decided by dividing the targeted geospatial space into fixed-size grid cells. The grid size is set to 10km $\times$ 10km ($R = 8 \times 7 = 56$) and 5km $\times$ 5km ($R = 16 \times 13 = 208$), and time interval is 60 minutes ($N = 24$). As the noise distribution, we use Gaussian distribution $p_{ti}(y_{ti}|n_{ti}) \propto \exp\left(-(y_{ti} - n_{ti})^2\right)$. We construct the potential $\phi_{tij} = \exp\left(-\operatorname{dist}(i,j)\right)$, where $\operatorname{dist}(i,j)$ is the Euclidean distance between the centers of cell $i$ and cell $j$ in the grid space. We create 10 instances by random sampling and averaged the attained objective function values for each setting. We also evaluate the estimation accuracy of the edge contingency table $(n_{tij})_{t \in [N-1], i, j \in [R]}$. We use normalized absolute error (NAE) as the evaluation metric, which is defined as $\frac{\sum_{t \in [N-1]} \sum_{i \in [R]} \sum_{j \in [R]} \left| n_{tij}^{\mathrm{true}} - n_{tij}^{\mathrm{est}} \right|}{\sum_{t \in [N-1]} \sum_{i \in [R]} \sum_{j \in [R]} n_{tij}^{\mathrm{true}}}$, where $n_{tij}^{\mathrm{true}}$ is the true value and $n_{tij}^{\mathrm{est}}$ is the estimated value of the edge contingency table. In addition to Proposed (L) and NLBP, we also evaluate the estimation accuracy of NLBP (rounded), which is a method that rounds the output of NLBP to integer values. Note that we do not evaluate the objective function values of NLBP (rounded). This is because the output of NLBP (rounded) completely violates the constraints on summation in the MAP inference problem (5) and the objective function value cannot evaluate whether the optimization problem is successfully solved or not.

---

[1]We use the data collected by the smartphone car navigation application of NAVITIME JAPAN Co., Ltd. (http://corporate.navitime.co.jp/en/). The data are collected with consent and appropriately anonymized.

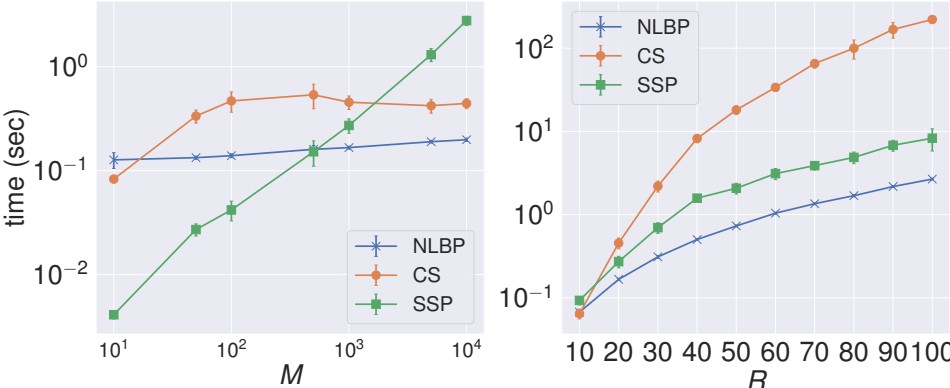

Figure 3: The computation time of each algorithm. The values are averages of 10 synthetic instances when $R$ is fixed to 20 (left) an $M$ is fixed to $10^3$ (right). $N$ is set 5 and the uniform potential is used.

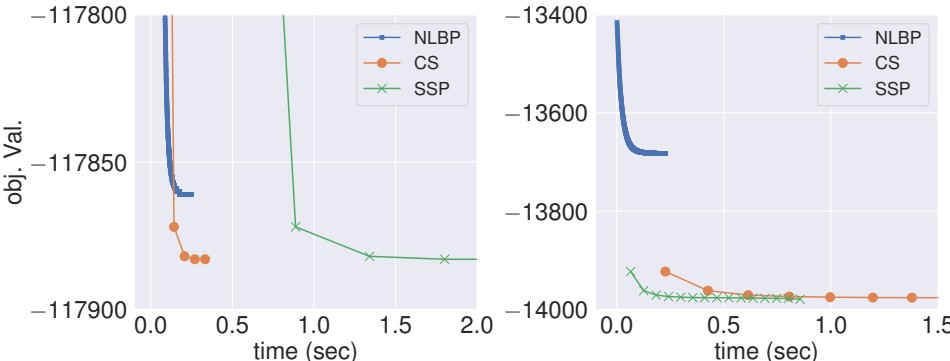

Figure 4: The relationship between running time and objective function value. The left figure shows the result of an instance of $R = 20, M = 10^4$ and uniform potential, and right figure shows that of $R = 30, M = 10^3$ and uniform potential.

Table 2 shows the results. We observe that Proposed (L) consistently attain smaller objective values and NAEs than the existing method. The superiority of the proposed method increase when $R$ is large and $M$ is small, and this is the same trend as the results with synthetic data. The NAE of NLBP is relatively large; this is due to the fact that small values are assigned to the elements of the output that should be 0, which is the same phenomenon seen in Figure 2. The NAE values are improved to some extent by rounding, but the proposed method is still superior.

## 6  Related Work

### 6.1  MAP inference for CGMs

Several methods have been proposed for the MAP inference of CGMs, but most of them take the approach of solving the approximate problem [16], which is derived by applying Stirling's approximation and continuous relaxation. For example, the interior point method [16], projected gradient descent [23], message passing [19] and Sinkhorn-Knopp algorithm [18] have been used to solve the approximate problem. In particular, [12] proposes a method to use DCA to solve this approximate problem. Although this approach is similar to our proposal in that it uses DCA, the purpose of applying DCA is totally different: our focus is to solve the MAP inference problem without using any approximation or continuous relaxation.

One of the few exceptions is the method proposed in [3], which solves the original MAP inference problem directly without using approximation. Our method follows this line of research, but there are two major differences. First, their method can only be applied to CGM on a graph with two vertices, and thus applicability is very limited. Since our method is consistent with this method when applied

to CGM on a graph with two vertices, our method can be regarded as a generalization of their method. Second, their work assumes accurate observations and does not handle observation noise.

[15] solves related collective MAP inference problems on path graphs. The problems addressed in this paper are different from ours; their purpose is finding the most likely assignments of the entire variables for each individual, while our purpose is finding the most likely node and edge contingency tables. In their settings, non-linear terms in the log posterior probability vanish, and the MAP inference problem can be solved easily by linear optimization approaches.

## 6.2 Difference of Convex Algorithm (DCA)

DCA, which is sometimes called Convex Concave Procedure [25], is a framework to minimize a function expressed as the sum of a convex function and a concave function [6]. DCA was originally proposed as a method for optimization in continuous domains. DCA has been used in various machine learning fields, such as feature selection [7], reinforcement learning [14], support vector machines [24] and Boltzmann machines [13].

Several studies have applied DCA to discrete optimization problems. This line of research is sometimes called discrete DCA [8]. [5, 11] propose algorithms to minimize the sum of a submodular function and a supermodular function. This algorithm is generalized to yield the minimization of the sum of an M/L-convex function and an M/L-concave function [8], where M-convex function and L-convex function are classes of discrete convex functions [10]. Although our work is closely related to these studies, it is not part of them. This is because our problem can be regarded as the minimization of the sum of two M-convex functions and a separable concave function, and this is not included in the class of functions dealt with in [8] [2].

## 7 Conclusion

In this paper, we propose a non-approximate method to solve the MAP inference problem for CGMs on path graphs. Our algorithm is based on an application of DCA. In the algorithm, surrogate functions can be constructed in closed-form and minimized efficiently by C-MCFP algorithms. Experimental results show that our algorithm outperforms approximation-based methods in terms of quality of solutions.

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
