# A Derivation of (5)

Because

$$\nu_t = \begin{cases} 1 & \text{if } t = 1, N, \\ 2 & \text{otherwise} \end{cases}$$

holds on path graphs, we have

$$F(\boldsymbol{n}) = \frac{M!}{Z^M} \cdot \frac{\prod_{t=2}^{N-1} \prod_{i\in[R]} n_{ti}!}{\prod_{t=1}^{N-1} \prod_{i,j\in[R]} n_{tij}!} \cdot \prod_{t=1}^{N-1} \prod_{i,j\in[R]} \phi_{tij}^{n_{tij}}$$

from (2). This gives

$$\begin{aligned} &- \log F(\boldsymbol{n}) - \log \Pr(\boldsymbol{y}|\boldsymbol{n}) \\ =& - \log M! + M \log Z \\ &- \sum_{t=2}^{N-1} \sum_{i\in[R]} \log n_{ti}! + \sum_{t=1}^{N-1} \sum_{i,j\in[R]} \log n_{tij}! - \sum_{t=1}^{N-1} \sum_{i,j\in[R]} n_{tij} \log \phi_{tij} - \sum_{t=1}^{N} \sum_{i\in[R]} \log p_{ti}(y|n) \\ =& \sum_{t=1}^{N-1} \sum_{i,j\in[R]} f_{tij}(n_{tij}) + \sum_{t=2}^{N-1} \sum_{i\in[R]} g(n_{ti}) + \sum_{t=1}^{N} \sum_{i\in[R]} h_{ti}(n_{ti}) + C, \end{aligned}$$

where $C$ is a constant. We can verify easily that the feasible region of problem (5) is $\mathbb{L}_M^{\mathbb{Z}}$ defined in (3).

# B Proofs

## B.1 Proof of Proposition 1

*Proof.* The function $\log z!$ is a discrete convex function, since

$$\begin{aligned} &\log(z+2)! + \log z! - 2 \log(z+1)! \\ =& \log(z+2) - \log(z+1) \geq 0. \end{aligned}$$

This yields that $f_{tij}(z) = \log z! - z \cdot \log \phi_{tij}$ is a discrete convex function and $g(z) = -\log z!$ is a discrete concave function. Because a univariate continuous convex function is also a discrete convex function, $h_{ti}(z) = -\log[p_{ti}(y_{ti}|z)]$ is a discrete convex function from Assumption 1. $\square$

## B.2 Proof of Proposition 2

*Proof.* First, we show that

$$- \log(w!) + \alpha \cdot (z - w) \geq - \log(z!), \quad \forall z \in \mathbb{Z}_{\geq 0} \tag{6}$$

holds for arbitrary $w \in \mathbb{Z}_{\geq 0}$, when $-\log(w+1) \leq \alpha \leq -\log w$. When $z \geq w$,

$$- \log(w!) + \alpha \cdot (z - w) + \log(z!) = \sum_{k=w+1}^{z} (\alpha + \log k) \geq 0$$

holds because $\alpha + \log(w+1) \geq 0$. When $z < w$,

$$- \log(w!) + \alpha \cdot (z - w) + \log(z!) \sum_{k=z+1}^{w} (-\alpha - \log k) \geq 0$$

holds because $-\alpha - \log w \geq 0$. Thus, inequality (6) holds.

Substituting $w = n_{ti}^{(s)}$ in (6), we get $\bar{g}_{ti}^{(s)}(z) \geq g(z)$ for all $z \in \mathbb{Z}_{\geq 0}$. This yields

$$\mathcal{R}^{(s)}(\boldsymbol{n}) = \sum_{t=2}^{N-1} \sum_{i=1}^{R} \bar{g}_{ti}^{(s)}(n_{ti}) \geq \sum_{t=2}^{N-1} \sum_{i=1}^{R} g(n_{ti}) = \mathcal{R}(\boldsymbol{n}).$$

Furthermore, since $\bar{g}_{ti}^{(s)}(n_{ti}^{(s)}) = g(n_{ti}^{(s)})$ from simple calculation, we get $\bar{\mathcal{R}}^{(s)}(\boldsymbol{n}^{(s)}) = \mathcal{R}(\boldsymbol{n}^{(s)})$. $\square$

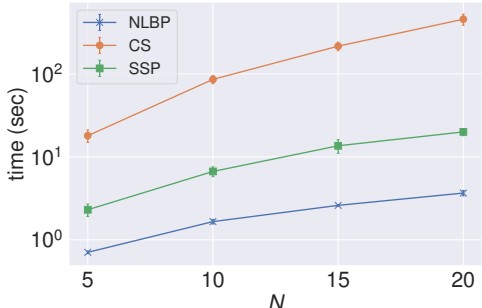

Figure 5: The computation time of each algorithm when varying $N$. The values are averages of 10 synthetic instances when $R$ is fixed to 50, $M$ is fixed to $10^3$ and the uniform potential is used.

### B.3 Proof of Proposition 3

*Proof.* There is a one-to-one correspondence between a feasible solution to the problem (5), $\boldsymbol{n}$, and a feasible solution to the MCFP instance constructed, $\boldsymbol{z}$, under the relationship $n_{ti} = z_{u_{t,i}w_{t,i}}$ and $n_{tij} = z_{w_{t,i}u_{t+1,j}}$; the constraint $\sum_{i \in [R]} n_{ti} = M$ is equivalent to the supply constraints at node $o$ and $d$, the constraint $\sum_{j \in [R]} n_{tij} = n_{ti}$ corresponds to the flow conservation rule at node $w_{t,i}$ and the constraint $\sum_{i \in [R]} n_{tij} = n_{i+1,j}$ corresponds to the flow conservation rule at node $u_{t+1,j}$. Moreover, corresponding $\boldsymbol{n}$ and $\boldsymbol{z}$ and have the same objective function value in problem (5) and the MCFP instance, respectively. These facts yield that $\boldsymbol{n}^*$ is the optimum solution of the problem (5). Because all the cost functions are discrete convex (this can be easily verified by Proposition 1 and definition of $\bar{g}_{ti}^{(s)}(z)$ in Proposition 2), the constructed instance belongs to C-MCFP. $\square$

## C  Additional experimental results

### C.1  Computation time

We run an experiment to investigate the relationship between $N$ (i. e. the number of vertices of the underlying graph) and the computation time. The results are shown in Figure 5. The values are averages of 10 synthetic instances when $R$ is fixed to 50, $M$ is fixed to $10^3$ and the uniform potential is used.

### C.2  Characteristics of the solutions

We run the same experiments as Figure 2 varying the value of $M$. The results are shown in Figures 6 and 7. The outputs of the proposed method and NLBP are totally different when $M$ is small, and they get closer as $M$ increases. This is owing to the nature of Stirling's approximation $\log x! \approx x \log x - x$; it is inaccurate especially when $x$ is small.

### C.3  Histogram interpolation

As an application of MAP inference of CGMs on path graphs, we can interpolate the time series of histograms between given two histograms. In this section, we show experimental results on this application and discuss the differences between the output of the proposed method and that of the existing method.

#### C.3.1  Settings

First, we briefly explain how to realize interpolation between two histograms by MAP inference of CGMs on path graphs. Suppose we are given histogram $\boldsymbol{\eta}_1 := [\eta_{11}, \ldots, \eta_{1R}]$ at time 1 and the histogram $\boldsymbol{\eta}_N := [\eta_{N1}, \ldots, \eta_{NR}]$ at time $N$. The interpolated histogram $\boldsymbol{\eta}_t$ at time $t (= 2, \ldots, N-1)$ is calculated by the following procedure.

1. Consider a CGM on a path graph with $N$ vertices.

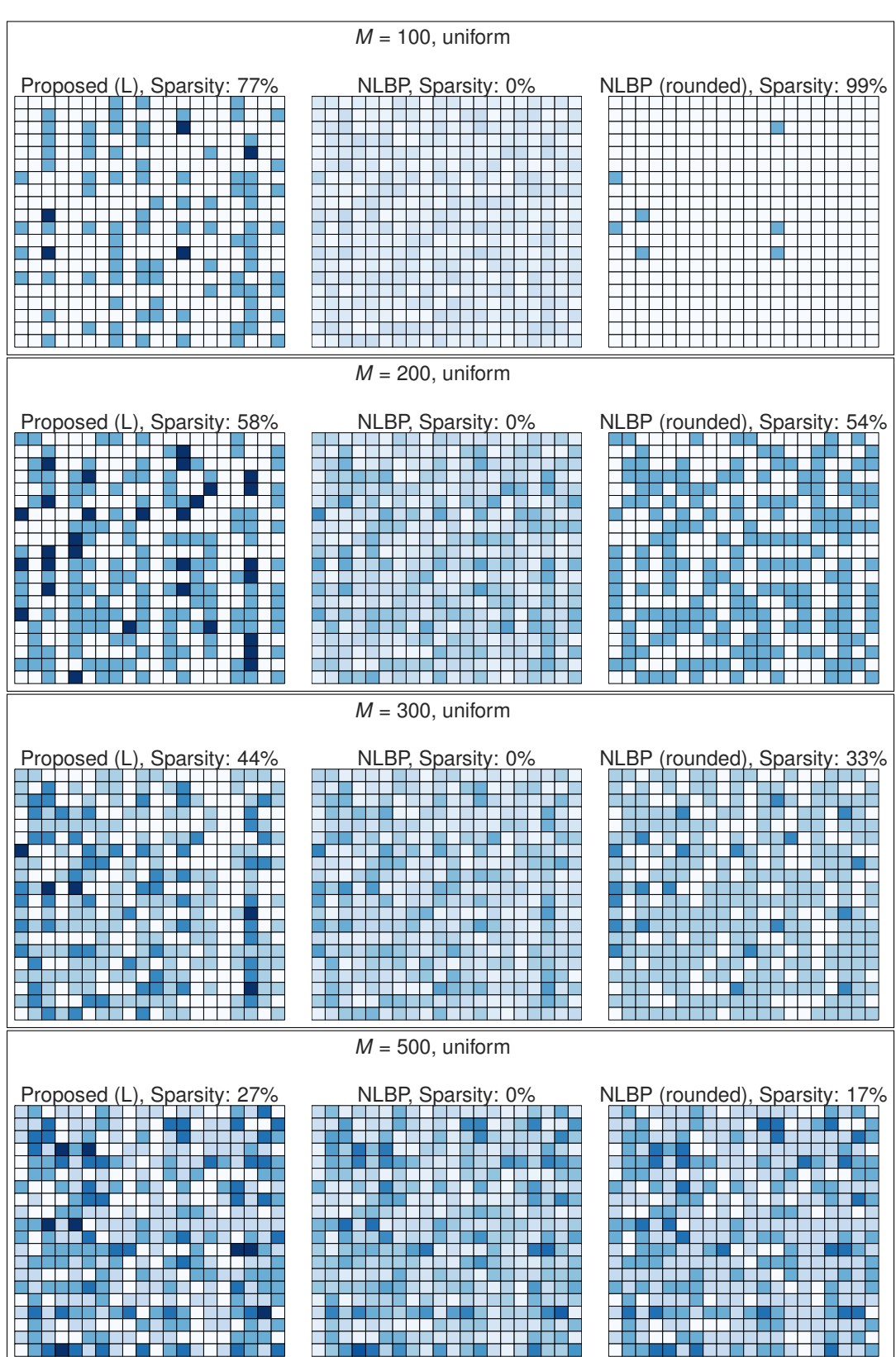

Figure 6: Comparison of solutions yielded by the proposed method (L), NLBP, and NLBP (rounded) when $R = 20$ and the uniform potential is used.

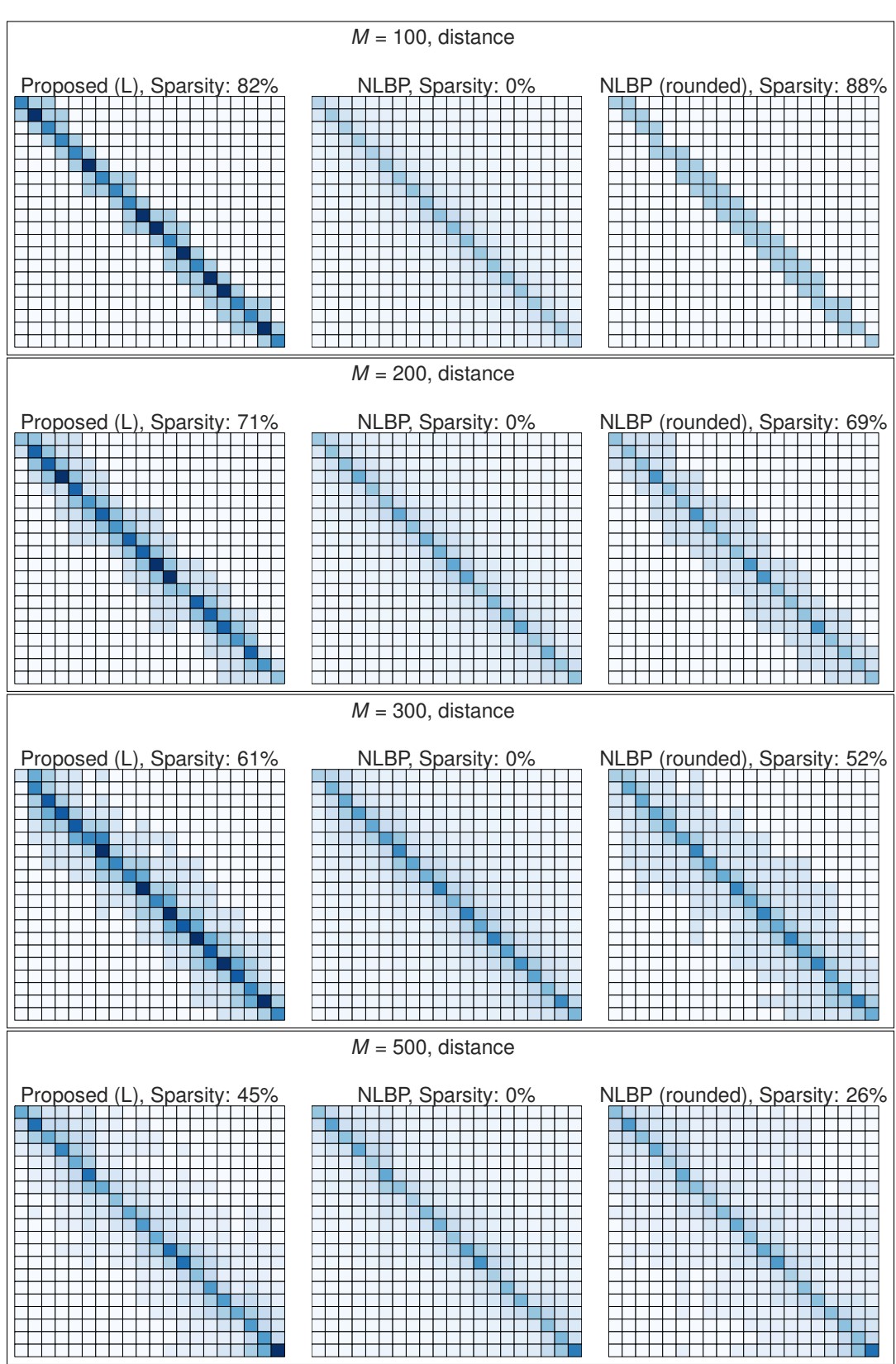

Figure 7: Comparison of solutions yielded by the proposed method (L), NLBP and NLBP (rounded) when $R = 20$ and the distance potential is used.

2. Let $\boldsymbol{y}_1 = \boldsymbol{\eta}_1$ and $\boldsymbol{y}_N = \boldsymbol{\eta}_N$.

3. $\boldsymbol{y}_t$ $(t = 2, \ldots, N-1)$ is treated as a *missing value*. This can be achieved by setting $h_{ti}(z) = 0$ $(t = 2, \ldots, N-1, \ i \in [R])$ in the objective function of the problem (5).

4. Find a solution $\boldsymbol{n}^*$ to the MAP inference problem under an appropriate potential $\phi$.

5. Obtain an interpolation result by $\eta_{ti} = n_{ti}^*$ $(t = 2, \ldots, N-1, \ i \in [R])$.

In our experiment, we consider a grid space of size $5 \times 5 = 25$ $(= R)$ and a histogram $\boldsymbol{\eta} := [\eta_1, \ldots, \eta_R]$ with a value $\eta_i$ for each cell $i$ $(= 1, \ldots, R)$. To get interpolation results which consider the geometric structure defined by Euclidean distance in the grid space, we set the potential $\phi_{tij} = \exp(-(r_i - r_j)^2 - (c_i - c_j)^2))$, where $(r_i, c_i)$ is the two-dimensional coordinate of the center of cell $i$ in the grid space. We set $N = 6$ and use Gaussian distribution $p_{ti}(y_{ti}|n_{ti}) \propto \exp(-5(y_{ti} - n_{ti})^2)$ for the noise distributions at $t = 1, N$.

### C.3.2 Results

The results are shown in Figure 8. Note that Figure 8 illustrates different objects from what is shown in Figures 2, 6 and 7; Figure 8 illustrates the interpolated node contingency table values $n_{ti}$ as two-dimensional grid spaces, while Figures 6 and 7 illustrate edge contingency table values $n_{tij}$ as matrices. As shown in the figure, NLBP tends to assign non-zero values to many cells, while proposed (L) assigns non-zero values to a small number of cells, resulting in sparse solutions. Moreover, the outputs of the proposed (L) are integer-valued while those of NLBP are not. This characteristic of the proposed method is beneficial for interpretability when the histogram values are the numbers of countable objects (e.g., the number of people in the area).

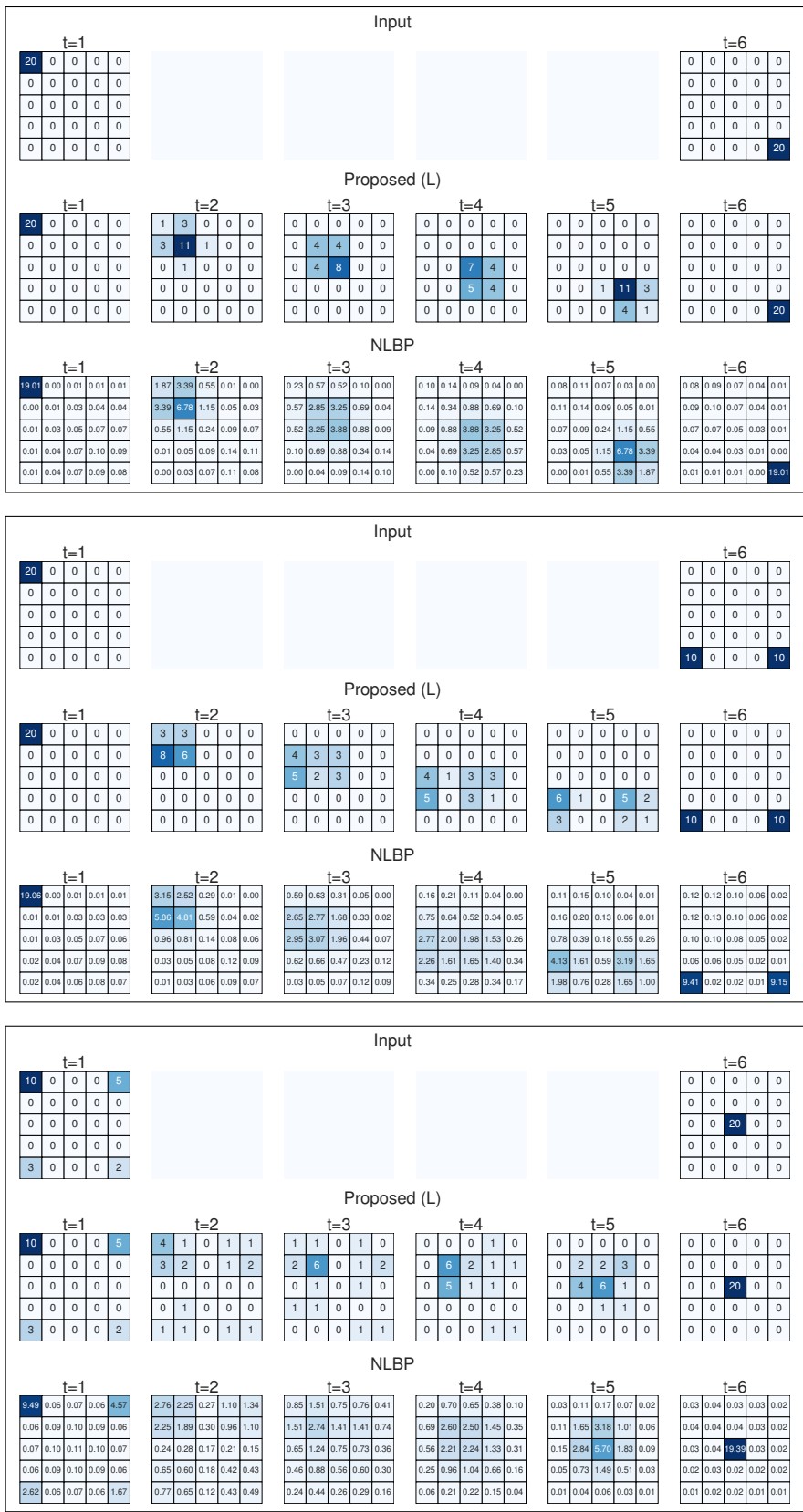

Figure 8: Three examples of interpolation results yielded by each method. In each example, three sequences of histograms in the two-dimensional grid space are presented; the first row shows the input histograms $\boldsymbol{\eta}_1$ and $\boldsymbol{\eta}_N$, the second row shows the interpolation results obtained by proposed (L), and the third row shows the interpolation results obtained by NLBP.