# OpenReview forum: "Non-approximate Inference for Collective Graphical Models on Path Graphs via Discrete Difference of Convex Algorithm"
_NeurIPS.cc/2021/Conference — NeurIPS 2021 Poster_

### Official Review · Reviewer_fJvy · 2021-07-01

**Rating:** 6
**Confidence:** 4

**Summary:**

The authors consider the problem of MAP estimate of count data given noisy count data. For example $X_1(t), X_2(t), \ldots, X_M(t)$ describe the location of an agent at time $t$, $N_i(t)$ is the number of agents at location $i$ at time $t$, and $Y_i(t)$ is a noisy estimate of $N_i(t)$. The goal is to estimate $N$ given $Y$. The noise distribution is assumed to be log-convex.

The authors develop an approach based on the difference of convex algorithm, which is an extension of the convex-concave procedure. They show that this procedure can monotonically improve a solution by repeatedly solving the minimum convex flow problem, which is known to be efficiently solvable.

The authors demonstrate that their approach improves over non-linear belief propagation with a continuous approximation of the discrete time problem.

**Limitations And Societal Impact:**

This work can be used for population tracking. However I am not too concerned about this paper increasing the negative aspects of this, as there are already methods that can solve the problem considered.

**Main Review:**

Pros:

The paper uses several interesting techniques to solve the problem of MAP estimate of count data on path graphs. I found the application of DCA and the use of min-cost convex flow to be interesting. They are able to work with the discrete problem, rather than previous work which needed to do a continuous relaxation.

The paper is mostly clear, however the notation is somewhat dense. This can be improved by adding names to the key variables. For example, instead of just $M$ the authors can write "the number of samples $M$", $\mathcal{Q}$ can be "the convex term $\mathcal{Q}$", $\bar{\mathcal{R}}$ can be "the proxy concave term $\bar{\mathcal{R}}$", etc. This will give the reader a reminder of what the variable represents.

Cons:

The algorithm presented is a heuristic without much analysis. It can be seen as somewhat incremental over [12], which also uses DCA.

The authors could compare to prior work on the bird migration benchmark, as used in [12]. Additionally, the experiments considered are all small scale and can be solved in less than a second. Can larger experiments be completed? This should be possible, at least for the synthetic experiments.

Questions:

Where would the noise on the count data come from in practice? To me, it seems more likely that there is noise on the positions $X_j(t)$. The authors should

**Time Spent Reviewing:**

3

---

> ### Author Response · Authors · 2021-08-09
> **Response to Reviewer fJvy**
>
> Thank you for your insightful comments and suggestions. Please find our responses to your questions and comments below.
>
> **[comment 1]
> The paper is mostly clear, however the notation is somewhat dense.**
>
> [response 1]
> Thank you for your suggestion. We will add names to key variables in the final version.
>
> **[comment 2]
> It can be seen as somewhat incremental over [12], which also uses DCA.**
>
> [response 2]
> As mentioned in Section 6.1, although the proposed method and the method in [12] share some similarities in terms of the use of DCA, their objectives are completely different; in [12], DCA is used to solve the *continuous* optimization problem after approximation, while in our paper, DCA is used to solve the *discrete* optimization problem without approximation.
>
> Furthermore, our work differs from [12] in that there is a difficulty in the step of minimizing surrogate function after applying DCA. We have successfully addressed this by reducing it to the minimum cost convex flow problem, and this is an important contribution of our paper.
>
> For these reasons, we believe that the difference between our work and [12] is not incremental.
>
> **[comment 3]
> Additionally, the experiments considered are all small scale and can be solved in less than a second. Can larger experiments be completed?**
>
> [response 3]
> As you mentioned, larger-scale experiments are possible with synthetic data. We have done additional experiments to investigate the performance on larger problems. The results are shown below. The experimental setup is the same as in Table 1. The average objective values and running times (seconds) of random 10 synthetic instances are shown with standard deviations in parentheses.
>
> 1. N = 5, **R=100**, M = 10^3, uniform potential
>     - objective value
>     NLBP: -1.15 * 10^4 (± 1.21 * 10^1)
>     SSP: -1.48 * 10^4 (± 1.86 * 10^1)
>     CS: -1.48 * 10^4 (± 1.97 * 10^1)
>     - computation time
>     NLBP: 2.57 (± 0.0112)
>     SSP: 8.46 (± 1.74)
>     CS: 226 (± 51.3)
> 2. N = 5, **R = 200**, M = 10^3, uniform potential (CS took more than 1000 seconds, so it is omitted in this case)
>     - objective value
>     NLBP: -9.83 * 10^3 (± 3.04 * 10^0)
>     SSP: -1.48 * 10^4 (± 2.48 * 10^1)
>     - computation time
>     NLBP: 9.75 (± 0.650)
>     SSP: 52.7 (± 12.1)
> 3. **N = 10**, R = 50, M = 10^3, uniform potential
>     - objective value
>     NLBP: -3.25 * 10^4 (± 3.39 * 10^1)
>     SSP: -3.55 * 10^4 (± 3.78 * 10^1)
>     CS: -3.55 * 10^4 (± 3.71 * 10^1)
>     - computation time
>     NLBP: 1.67 (± 0.151)
>     SSP: 6.58 (± 0.812)
>     CS: 89.5 (± 9.43)
> 4. **N = 20**, R = 50, M = 10^3, uniform potential
>     - objective value
>     NLBP: -7.09 * 10^4 (±  4.48 * 10^1)
>     SSP: -7.73 * 10^4 (± 6.13 * 10^1)
>     CS: -7.73 * 10^4 (± 6.13 * 10^1)
>     - computation time
>     NLBP: 3.70 (± 0.200)
>     SSP: 20.3 (± 3.14)
>     CS: 404 (± 62.8)
>
> This information will be compiled in the form of figures and tables and added to the final version.
>
> **[comment 4]
> Where would the noise on the count data come from in practice?**
>
> [response 4]
> The comment of the reviewer on OpenReview is cut off in the middle (After "The authors should ..." is missing), so we may not have fully understood the reviewer's intent. We understood the question to be as follows:
>
> "There are two possible processes for generating aggregated data: (a) individual positions are determined → individual values are aggregated → noise is added to the aggregated values (b) individual positions are determined → noise is added to the individual positions → individual values are aggregated. Although (a) is used in this paper, isn't (b) the correct one in reality?"
>
> In the example of Mobile Spatial Statistics [22] given in the introduction, (a) is correct because noise is added after the aggregation to prevent the identification of personal mobility information. Also, in the case of measuring the number of people at each location using sensors and cameras in a traffic network, (a) is more realistic because counting errors may occur in each device. However, in the animal migration model used in previous studies, (b) may be correct depending on the data collection method and the characteristics of the observation device. In the existing CGM studies, (a) is mainly used and this paper follows it, but (b) is sometimes used (e.g. [18]). It is important to use appropriate models depending on the situation.

---

> > ### Comment · Reviewer_fJvy · 2021-09-01
> > **Thank You**
> >
> > Thank you for your clarifications and experiments. I decided to raise my score to a 6.

---

### Official Review · Reviewer_2Z5W · 2021-07-16

**Rating:** 7
**Confidence:** 4

**Summary:**

The paper proposes a non-approximate method for solving MAP inference problem in collective graphical model (CGM). The paper uses difference of convex algorithm (DCA) strategy as a solver for the optimization problem since the objective function can be expressed as a sum of convex and concave function. The solution is obtained via minimization of a surrogate function that upper bounds the objective function. The technical details appear to be correct and the simulation results show the superiority of the proposed algorithm over its competitors. Further, a real world application involving car trajectories is performed to validate the performance of the proposed methodology.

**Ethical Concerns:**

None.

**Limitations And Societal Impact:**

Nothing as such.

**Main Review:**

The contribution appears to be technically correct and is a valid addition to the graphical model literature. Although the DCA has been used in other contexts its application to solve the MAP inference in CGM is novel. The algorithm also not too much worse compared to its competitors in terms of scalability. The paper is overall well-written although at some places (especially in the section where minimum cost flow algorithm was described) a bit more clarity would have been ideal. I have a couple of minor comments for the author(s)

- How does the choice of the surrogate function in DCA affect the MAP inference?

-How does one choose the value of M in the c-MCFP algorithm?

-Is there any practical guideline when to use the successive shortest path (SSP) or the capacity scaling (CS)?


******* Change after rebuttal **********

I am satisfied with the response from the author(s). The rebuttal has answered most of the queries related to the technical details of the paper. It also has also clarified some of the questions regarding the experiments with some additional results. I am keeping my previous score "Good paper, accept" unchanged.

**Time Spent Reviewing:**

3 hours

---

> ### Author Response · Authors · 2021-08-09
> **Response to Reviewer 2Z5W**
>
> Thank you for your constructive feedback. Please find our responses to your questions and comments below.
>
> **[comment 1]
> The paper is overall well-written although at some places (especially in the section where minimum cost flow algorithm was described) a bit more clarity would have been ideal.**
>
> [response 1]
> Thank you for your suggestion. For the part on the minimum cost flow algorithm, we will add the content of [response 5] to Reviewer 6fXy for intuitive explanations of the algorithms and detailed computational analyses in the final version.
>
> **[comment 2]
> How does the choice of the surrogate function in DCA affect the MAP inference?**
>
> [response 2]
> We found experimentally that the output of Proposed (L) has a larger variance in the count at the vertices n_{ti} than the output of Proposed (R). This may be because Proposed (L) tends to "estimate large values larger and small values smaller" than Proposed (R). However, as can be seen from the results in Table 1, there is no significant difference in the log-likelihood achieved by each algorithm. Thus, the effect of the choice of the surrogate function on the MAP inference is not considered to be significant.
>
> **[comment 3]
> How does one choose the value of M in the C-MCFP algorithm?**
>
> [response 3]
> In most of the existing studies on MAP inference of CGM, the value of M is assumed to be given. In this paper, we follow them and treat M as given.
>
> However, it is an important problem to determine M appropriately in real-world applications. In fact, our method can be extended to the setting where M is not given; in the minimum cost flow algorithm in each iteration, we monitor the objective function value while successively pushing a unit flow from vertex o to vertex d, and stop when the objective function value reaches a minimum. We can show that this method works well because the objective function value is a convex function with respect to the flow amount. If necessary, we will add a short section describing this method to the Appendix.
>
> **[comment 4]
> Is there any practical guideline when to use the successive shortest path (SSP) or the capacity scaling (CS)?**
>
> [response 4]
> Basically, as shown in the theoretical computational analysis (Section 4.3) and experimental results (Figure 3), it is recommended to use SSP when R and N are large, and CS when M is large. We conducted additional experiments on computation time to make our decision criteria clearer. The result is that SSP is better to use when roughly M/logM ≤ NR^2 holds, and CS is better to use otherwise. This result is consistent with the results of the theoretical computational analysis.

---

### Official Review · Reviewer_6fXy · 2021-07-18

**Rating:** 6
**Confidence:** 4

**Summary:**

The paper proposes a new method of estimating MAP solutions from a collective graphical model.

**Limitations And Societal Impact:**

No potential negative societal impact is detected.

**Main Review:**

The proposed method is very solid. Solving the MAP problem in the discrete space does not need approximating the log joint function as in continuous approximations.

The performance of the proposed method outperforms the baseline.

The writing is generally clear.

The work also has a few limitations.
1. The proposed method only applies to a small scope of problems. With the few characterizations (CGM, path-graph, and small counts), the proposed method can only show strength in a small number of problems.

2. The experiment on the real data uses true counts as the ground-truth, but this may not be correct. With the model specification in the experiment, true counts may not be the MAP solution.

Small issues:
Is it really useful to differentiate the three algorithms L, M, and R given their similarity? If so, can the authors put more analysis of how differences arise from the three configurations?

I am not sure the two propositions make the writing clear. They are only analyses of the problem in focus. The notations are from the previous text, so the propositions do not stand by themselves.

A little more insight into C-MCFP can assist the understanding. I have a rough idea that one can get a good solution with (even greedy) descending steps. But can you indicate the complexity here?



**Time Spent Reviewing:**

2

---

> ### Author Response · Authors · 2021-08-09
> **Response to Reviewer 6fXy**
>
> Thank you for your valuable feedback. Please find our responses to your questions and comments below.
>
> **[comment 1]
> The proposed method only applies to a small scope of problems.**
>
> [response 1]
>  As you say, there are limitations in the scope of our method. However, we believe that the scope of our method is particularly important in the field of CGM research. There are two reasons for this:
>
> 1. As mentioned in the second paragraph of the Introduction, CGMs on path graphs can handle Markov models, which can represent time-series data including human and animal migration, and have been used in many previous studies.
>
> 2. In real-world applications, it is expected that the count values (i.e., n_{ti} and n_{tij}) will frequently become small. For example, human movements estimation presented in the second paragraph of Section 3.2, it is desirable to divide the area as small as possible in order to obtain more detailed information for real-world applications such as traffic management and urban planning. If the areas are divided into smaller ones, the number of areas R becomes larger, while the total population M does not change, so each count value becomes close to zero. Algorithms that work well in such situations are considered to be of high importance.
>
> **[comment 2]
> The true counts may not be the MAP solution in the real data experiments.**
>
> [response 2]
> As you mention, the true count is not always the MAP inference solution. In order to evaluate the algorithm from multiple perspectives, we utilize two metrics: the log-likelihood to evaluate the performance of MAP inference, and the normalized absolute error (NAE) from the true count to evaluate the practical estimation performance. By using these two metrics together, we believe that we have evaluated the method appropriately as long as possible, although it may not be perfect.
>
> **[comment 3]
> Is it really useful to differentiate the three algorithms L, M, and R given their similarity?**
>
> [response 3]
> We apologize for the confusion. The reason why we evaluate each of Proposed (L), Proposed (M), and Proposed (R) is not that we need to distinguish them, but rather to show that the proposed method performs robustly regardless of the choice of the surrogate function. The experimental results show that there is no significant difference between these methods, although Proposed (L) is slightly better. This indicates that the proposed method is robust with respect to the choice of hyperparameters \alpha_{ti}^{(s)}. In the final version, we emphasize this.
>
> **[comment 4]
> I am not sure the two propositions make the writing clear.**
>
> [response 4]
> Thank you for pointing this out. It may be more appropriate to treat Propositions 1 and 2 as Observation rather than Proposition. We will fix this in the final version.
>
> **[comment 5]
> A little more insight into C-MCFP can assist the understanding.**
>
> [response 5]
> We provide intuitive explanations of the C-MCFP algorithms (SSP and CS) and analyses of the computational complexity. We plan to add these contents to the final version.
>
> 1. SSP is an algorithm that successively augments unit flow along the shortest path from a supply node (i.e. b_i > 0) to a demand node (i.e. b_i < 0) in a *residual graph*, which is an auxiliary graph calculated from the current flow.
> Given a C-MCFP instance with graph G = (V, E), the shortest path in the residual graph can be computed in O(|E| log |V|) by Dijkstra's algorithm with a binary heap, and the augmentation of the flow can be done in O(|E|). The augmentation is performed O(B) times totally, where B := (\sum_{i \in V} |b_i|)/2, so the total time complexity is O(B |E| log |V|).
>
> 2. CS resembles SSP, but it differs in that it tries to push a large amount of flow, rather than a unit amount of flow, in a single augmentation. In CS, the number of shortest path calculations and flow augmentations can be bounded O(|E| log U) times, where U := \max_{i \in V} |b_i|) so the total computational complexity is O(|E|^2 log |V| log U).

---

> > ### Comment · Reviewer_6fXy · 2021-08-31
> > **Thank you for your responses**
> >
> > Your responses address most of my concerns -- thank you!

---

### Decision · Program_Chairs · 2021-09-27

**Decision:**

Accept (Poster)

**Comment:**

The authors present a new algorithm for MAP inference in chain-structured Collective Graphical Models that uses flow techniques to solve the problem in discrete space, as opposed to previous works which use continuous relaxations. Three reviewers found the techniques to be correct and represent an advance over prior methods, with sufficient evidence presented to show the advantages of working in discrete space. The scores were 6, 6, 7. One reviewer raised their score after the author response. The meta-reviewer recommends accept.